# AGENT WORKFLOW MEMORY

## ABSTRACT

Despite the potential of language model-based agents to solve real-world tasks such as web navigation, current methods still struggle with long-horizon tasks with complex action trajectories. In contrast, humans can flexibly solve complex tasks by learning reusable task workflows from past experiences and using them to guide future actions. To build agents that can similarly benefit from this process, we introduce Agent Workflow Memory (AWM), a method for inducing commonly reused routines, i.e., *workflows*, and selectively providing workflows to the agent to guide subsequent generations. AWM flexibly applies to both *offline* and *online* scenarios, where agents induce workflows from training examples beforehand or from test queries on the fly. We experiment on two major web navigation benchmarks — Mind2Web and WebArena — that collectively cover 1000+ tasks from 200+ domains across travel, shopping, and social media, among others. AWM substantially improves the baseline results by 24.6% and 51.1% relative success rate on Mind2Web and WebArena while reducing the number of steps taken to solve WebArena tasks successfully. Furthermore, online AWM robustly generalizes in cross-task, website, and domain evaluations, surpassing baselines from 8.9 to 14.0 absolute points as train-test task distribution gaps widen.

## 1 INTRODUCTION

Language model (LM) based agents are rapidly improving, and are now able to tackle digital tasks such as navigating the web (Zhou et al., 2024; Deng et al., 2023) or operating mobile apps (Rawles et al., 2023; 2024). Current agents mostly integrate a fixed set of given examples via training (Fu et al., 2024; Murty et al., 2024) or in-context learning (Zheng et al., 2024). This allows them to perform well on action sequences similar to those presented in these examples, but results in a lack of robustness to changes in task contexts or environments (Deng et al., 2023). Essentially, they fail to grasp the key to disentangling increasingly complex tasks — to extract and learn reusable task workflows shared across similar tasks and environments (Yu et al., 2023; Wang et al., 2024a). Moreover, as agents solve each task separately, they do not learn from past successes and failures, and are therefore unable to adapt over time (Yoran et al., 2024).

Motivated by how humans abstract common task routines from past experiences and apply such knowledge to guide future activities (Chi et al., 1981; 2014), we propose agent workflow memory (AWM) (§2) to realize a similar mechanism in agents. AWM induces workflows from agent trajectories by extracting reusable routines, and then integrates these workflows into agent memory to guide future task-solving processes. Each workflow represents a goal with a common routine extracted from available action trajectories, which allows it to effectively capture the most essential and reusable skills agents need to acquire to successfully solve increasingly complex tasks. As an example, Figure 1 shows workflows induced by AWM on the WebArena map test split of the benchmark (Zhou et al., 2024). AWM starts with a basic set of built-in actions and

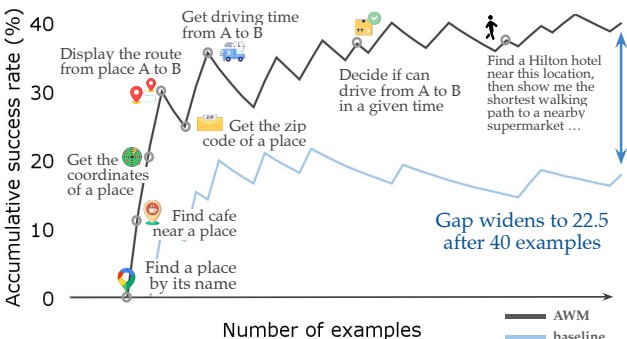

Figure 1: AWM enables agents to continuously induce and apply workflows to improve performance, compared to stagnant baselines. We show results by AWM on the WebArena map split as an example.

solves new tasks in a streaming manner, continuously inducing workflows from the task at hand, e.g., learning to "find a place by its name" from the first few examples. Moreover, AWM continues to build more complex workflows on top of new experiences and previously acquired workflows. For example, the "find a place by its name" workflow, once induced, effectively serves as a subgoal to build a more complex workflow "get the zip code of a place." Such continual learning mechanisms create a snowball effect to induce and apply increasingly complex workflows while expanding the agent memory, often yielding a substantial performance gap over a vanilla agent that does not adapt. This gap over the baseline rises as high as 22.5 points on WebArena after rolling over only tens of examples (as shown by Figure 1).

AWM readily operates in both *offline* and *online* scenarios, where annotated examples are either available or non-existent. When high-quality annotated examples are available for a task, AWM operating in an *offline* fashion can extract reusable workflows from these canonical examples and integrate them into memory to assist test-time inference. Even if no annotated examples exist, AWM *online* can also run in an *supervision-free setting*, where it iteratively induces workflows from self-generated past predictions that are judged correct by an evaluator module.

We evaluate AWM on two agent web navigation benchmarks (§3): WebArena, which provides rigorous execution-based evaluation (Zhou et al., 2024), and Mind2Web, which emphasizes broad tasks and domain coverage (Deng et al., 2023). On WebArena, AWM improves over the top published autonomous method (Drouin et al., 2024) by $51.1\%$ relative success rate, and even outperforms methods augmented with human expert written workflows (Sodhi et al., 2023) by $7.9\%$. On Mind2Web, AWM effectively improves the cross-task results by $24.6\%$ in relative step-wise success rate.

We further demonstrate the generalizability of AWM on both datasets. On WebArena, we create a cross-template subset where each example is instantiated from different task templates. AWM still consistently surpasses all baseline approaches, demonstrating its reliable cross-task workflow adaptability (§3.1). On Mind2Web, we evaluate AWM on the cross-website and cross-domain test splits to examine its domain generality, where it scores $8.9$–$14.0$ absolute points higher over baseline, and the margins become more substantial as the train-test distribution gap widens (§3.2). Both results show the superior generalization of AWM across tasks, websites, and domains.

## 2 AGENT WORKFLOW MEMORY

In this section, we first describe the web navigation task (§2.1), then introduce the workflow representation (§2.2), and describe the mechanism of AWM as well as various instantiations (§2.3).

### 2.1 PROBLEM STATEMENT

For the purpose of this paper, we consider agents with a language model backbone $L$ and text-based memory $M$, where the base memory contains documentation of built-in actions such as CLICK and TYPE.[1] To solve a task specified by a natural language (NL) instruction $q$, the agent acts in an environment defined by a transition function $T$. For each time step $t_i$, the environment state $s_i$ gives observation $o_i$, which is then passed into the model to generate action $a_i$ via $L(q, M, o_i) \rightarrow a_i$. The action is executed in the environment and changes the state as $T(s_i, a_i) \rightarrow s_{i+1}$. This observe-act loop iterates until the model predicts the stop action $a_i$ =STOP, or reaches a task termination condition, e.g., a maximum pre-determined number of steps.

Each completed task forms an experience $e$, which comprises an NL instruction $q$ and a trajectory of steps attempting to solve the task, where each step $p$ contains the agent observation $o$ obtained from the current state, and action taken by the agent $a$, formulated as $p = (o, a)$. Our goal is to induce useful workflows $\mathcal{W} = \{w\}$ from the set of experiences $\mathcal{E} = \{e\}$ constructed from past or collected examples, using an induction module $I$ via $I(\mathcal{E}) \rightarrow \mathcal{W}$. We add induced workflows into the agent memory $M$ as guidance for subsequent task-solving.

Next, we introduce the workflow representation design, workflow induction method, and agent memory update with workflows in varied setups.

---

[1]Memory is usually implemented as a system prompt or auxiliary information in the main prompt context.

## 2.2 WORKFLOW REPRESENTATION

Similar to an experience, a workflow comprises two components: first, a textual description of the workflow $d$; and second, a series of steps to finish the workflow $(p_1, p_2, \cdots)$, as shown in Figure 2.

**Workflow Description** To present workflows in a format where agents can learn from them properly, it is important to describe the high-level goal of the series of actions. Therefore, we associate each workflow with an NL task description $d$, essentially a summary of the workflow's function, by heuristically extracting from experience instructions or summarizing with an LM (see §2.3).

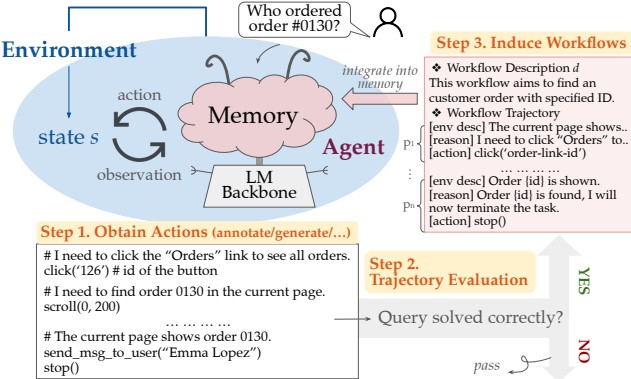

**Workflow Trajectory** The workflow trajectory contains a series of steps $(p_1, p_2, \cdots)$ to finish the process described in $d$. Each $p$ consists of three parts, demonstrated in $p_n$ in

Figure 2: Illustration of our AWM pipeline: an agent takes actions to solve given queries, induces workflows from successful ones, and integrates them into memory.

Figure 2, Step 3. (1) A description of the current environment state in NL, such as "Order {id} is shown"; (2) The reasoning process elaborated by the agent to decide which action to generate based on observations, such as "Order {id} is found, I will now terminate the task."; and (3) an action represented as an executable program over the environment, i.e., `stop()` that realizes termination.

## 2.3 INDUCING AND USING WORKFLOWS

At the core of AWM is an induction module $I$ that induces a set of workflows $\mathcal{W}$ from one or more past agent experiences $\mathcal{E} = \{e_i\}_{i=1}^m$. Each experience $e = (q, P^e)$ contains an NL task instruction $q$ and an action trajectory that consists of a sequence of steps (observation and action) $P^e = (p_1^e, ..., p_n^e)$ that were taken to solve $q$. The workflow induction module operates by taking in $\mathcal{E}$ and producing a set of workflows, as $I(\mathcal{E}) \rightarrow \mathcal{W} = \{w\} = \{(d_j, P_j^d)\}$.

**LM-based Workflow Induction** To produce workflows that more accurately capture reusable trajectories across tasks, we propose an LM-based module $I$ that prompts the agent to extract common sub-routines from one or more input experiences.

Different from task instructions that specify concrete, less-repetitive tasks, e.g., "Buy dry cat food on Amazon and deliver to my address", we deliberately prompt models to induce workflows at finer granularities, i.e., a sub-task "search for a product on Amazon" that frequently re-appears as part of multiple similar instructions. Meanwhile, instead of giving example-specific values (e.g., "dry cat food"), we enhance workflow generality by abstracting out example-specific contexts, i.e., replacing "dry cat food" with a more general name "{product-name}" by specifying this in the workflow induction prompts. These workflows are segmented (based on double-line breaks in the model output) and stored separately in the workflow memory. See §A for the model prompts, example workflows, and an examination of quality.[2]

After the workflows $\mathcal{W}$ are induced, they are then integrated into the agent as auxiliary memory, $M + \mathcal{W} \rightarrow M_w$, where $M$ stands for the original agent memory, and $M_w$ stands for the agent memory augmented with induced workflows. When solving a given instruction $q$, the agent now produces a series of actions by $L(q, M_w, o) = L(q, M + W, o) \rightarrow a$. In the following, we introduce AWM in use in two scenarios:

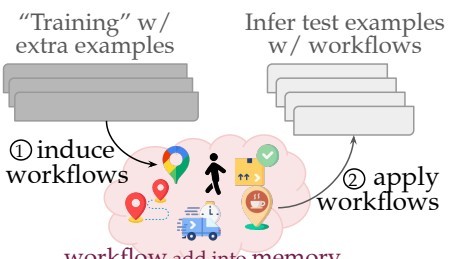

Figure 3: Illustration of $\text{AWM}_{offline}$.

**Offline Scenario** AWM can operate in an *offline* scenario when additional canonical experiences are available, such as data annotated by humans

---

[2]We also explore a rule-based workflow induction method. See §B for more detailed experiments.

or synthesized by models. In this case, we perform workflow induction and utilization in two standalone processes. As seen in Figure 3, AWM first takes in all training examples from a website by concatenating them into a single prompt, and feeds them to the LM to create a set of workflows at 'training' time; $I(\mathcal{E}_{train}) \rightarrow \mathcal{W}_{offline}$. Second, AWM incorporates all induced workflows into the agent memory at inference time to solve test instructions $L(q, M + \mathcal{W}_{offline}, o_i^{test}) \rightarrow a_i^{test}$. Since the workflows are fully induced before test-time inference, the agent uses the same workflow memory $\mathcal{W}_{offline}$ to solve each test.

**Online Scenario**   Extra canonical experiences are not always available or easy to collect, especially those that cover the same domains and tasks as the test instructions. AWM also works in an online, supervision-free setting, where only test queries are needed. Agents with $AWM_{online}$ process test queries in a streaming fashion, where the agents conduct the loop of induce, integrate, and utilize workflows after running inference for each test task.

Concretely, the agent starts with the default memory $M$; given the $t$-th test instruction $q_t$ passed into the agent, the agent attempts to solve the task by generating an action trajectory $(p_1^t, p_2^t, \cdots)$, which collectively forms an experience $e_t = (q^t, \{p^t\})$. We adopt the LM-based evaluation model of Pan et al. (2024) to output a binary label, $L_{eval}(e^t) \in \{0, 1\}$, that judges if $e^t$ successfully solves $q^t$ by prompting a neural model. If $e^t$ is predicted as success, i.e., 1, we then transform it into workflow(s) $I(e^t) \rightarrow \{w^t\}$

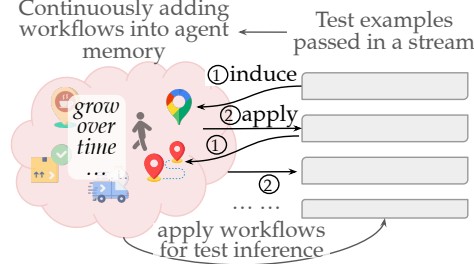

Figure 4: Illustrations of $AWM_{online}$.

and add $\{w^t\}$ into the agent memory $M^t + \{w^t\} \rightarrow M^{t+1}$, which serves as the agent memory to process the $t + 1$-th instruction. As depicted in Figure 4, we continue this memory-updating process by iteratively predicting actions for and inducing workflows from streamed test instructions, until all tests are processed. We evaluate the success rate of predicted action trajectories $\{p^t\}$ for all tests.

## 3  EXPERIMENTS

In this section, we experiment on two major web navigation benchmarks – WebArena (§3.1) and Mind2Web (§3.2). For each benchmark, we first introduce the benchmark and top-performing baseline methods, then present our AWM approach and showcase its ability to achieve superior task success and generalization across varied setups.

For both benchmarks, we conduct AWM on a website basis. In other words, we group examples by their associated websites, and respectively run AWM on each group. This mechanism maintains an small collection of workflows that are nonetheless relevent to the test tasks.

### 3.1  WEBARENA

WebArena (Zhou et al., 2024) provides 812 web navigation tasks on five websites that cover four common application domains: e-commerce, social forum discussions, collaborative software development, and content management. Most importantly, WebArena supports rigorous evaluation on the functional correctness of agent trajectories.

We adopt the current state-of-the-art method without human-annotated site-specific knowledge, BrowserGym (Drouin et al., 2024), which altered the agent default action space. We adopt the BrowserGym framework and its default action space, and represent webpages using accessibility trees, following the environment representation in Zhou et al. (2024). Because BrowserGym inputs both webpage HTML and accessibility tree representations, to keep a fair comparison with our method, we also run the BrowserGym version with only accessibility tree webpage representations, denoted as $BrowserGym_{ax-tree}$. We also compare to the SteP method (Sodhi et al., 2023), which uses 14 human expert written workflows tailored to solving WebArena. Our method, in contrast, uses no human supervision and is not tailored to the WebArena setting.

Following baseline approaches, we use GPT-4 (`gpt-4-0613`) with a temperature of $0.0$ to ensure mostly stable model outputs. Because WebArena only has test examples and no additional high-quality, domain-aligned examples exist, we only conduct AWM in the online setting as in §2.3.

Table 1: Task success rate on WebArena using `gpt-4`, and score breakdown on five website splits.

| Method | Total SR | Shopping | CMS | Reddit | GitLab | Maps | # Steps |
|---|---|---|---|---|---|---|---|
| *With human engineered workflows* | | | | | | | |
| *SteP (Sodhi et al., 2023) | 33.0 | **37.0** | 24.0 | **59.0** | **32.0** | 30.0 | - |
| *Autonomous agent only* | | | | | | | |
| WebArena (Zhou et al., 2024) | 14.9 | 14.0 | 11.0 | 6.0 | 15.0 | 16.0 | - |
| AutoEval (Pan et al., 2024) | 20.2 | 25.5 | 18.1 | 25.4 | 28.6 | 31.9 | 46.7 |
| BrowserGym (Drouin et al., 2024) | 23.5 | - | - | - | - | - | - |
| BrowserGym$_{ax-tree}$ | 15.0 | 17.2 | 14.8 | 20.2 | 19.0 | 25.5 | 7.9 |
| AWM (OURS) | 35.5 | 30.8 | 29.1 | 50.9 | 31.8 | **43.3** | **5.9** |

### 3.1.1 MAIN RESULTS

As shown in Table 1, our AWM achieves the best published results on WebArena, surpassing the BrowserGym baseline by $12.0$ absolute points and $51.1\%$ relative increase in overall success rate. Notably, AWM also outperforms SteP, which uses strong domain-specific supervision from human-written workflows, by a $7.6\%$ relative increase in overall success rate. According to the breakdown on five websites, our AWM method substantially enhances the agent performance across all websites over the BrowserGym baseline, by $11.8$–$30.7$ absolute points, indicating its general applicability across varied domains and tasks.

Beyond task success, we also evaluate the average number of steps the agent takes to solve a task, as shown in the rightmost column in Table 1. AWM conducts about $2.0$ fewer steps per example than the BrowserGym baseline. Further compared to the Autoeval (Pan et al., 2024) method, which necessitates additional evaluation and refinement steps to solve tasks correctly, our AWM approach uses $40.8$ fewer steps on average. Both comparisons show that AWM obtains high success rates while maintaining efficient trajectories.

### 3.1.2 EFFICIENT LEARNING FROM SMALL AMOUNTS OF DATA

To demonstrate the behavior of the AWM$_{online}$ method, we illustrate the cumulative success rate over the process of online evaluation, by evaluating the average success rate of the first $k$ finished examples.

As in Figure 5, the agent exhibits a fast learning curve in the beginning (between 0–40 examples), by acquiring the most essential workflows, which results in higher success rates. Afterward, agents learn more advanced workflows (Figure 1), while success rates gradually stabilize to the highest point in the early learning phase. This showcases AWM's efficient learning process, which substantially improves performance with merely tens of examples.

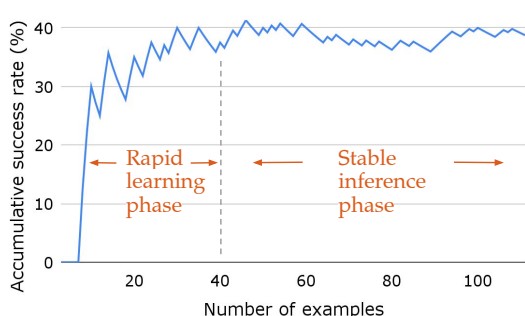

Figure 5: AWM enables rapid learning from a small amount of data, i.e., about 40 queries, using WebArena map test split as an example.

### 3.1.3 CROSS-TEMPLATE WORKFLOW GENERALIZATION

Some tasks in the WebArena benchmark have highly overlapping canonical trajectories, due to the benchmark construction process that instantiates multiple examples from a single underlying task template. AWM intuitively improves in-template success rate, that is, given one workflow induced from a successful example, it would be theoretically easier to solve all other examples generated from the same task template.

To confirm that the benefits of AWM are not just from learning workflows that help only within a template, and investigate whether AWM can obtain cross-template ($\approx$cross-task) generalization, we extract a subset of WebArena examples sourcing from non-overlapping templates, by grouping

Table 2: Task success rate on the cross-template subset of WebArena, as well as the result breakdown on each website split. We mark the number of examples for each website split under the name.

| Method | Total SR | Shopping (51) | CMS (45) | Reddit (24) | GitLab (45) | Maps (32) |
|---|---|---|---|---|---|---|
| *With human engineered workflows* | | | | | | |
| *SteP (Sodhi et al., 2023) | 32.1 | **26.5** | **29.3** | **52.2** | 27.3 | 36.4 |
| *Autonomous agent only* | | | | | | |
| AutoEval (Pan et al., 2024) | 23.2 | 12.2 | 17.1 | 21.7 | **31.8** | 36.4 |
| BrowserGym$_{ax-tree}$ | 20.5 | 10.4 | 17.8 | 23.1 | 27.3 | 28.6 |
| AWM (OURS) | **33.2** | 24.5 | **29.3** | **52.2** | **31.8** | **39.4** |

examples by their templates and randomly choosing one example from each template group. We run AWM on this cross-template subset and examine if it achieves similar performance gains.

As shown in Table 2, AWM still achieves the highest performance, overall and on each website split. These results demonstrate that AWM induced workflows can effectively generalize across different tasks, i.e., examples instantiated from different task templates.

**Building increasingly complex workflows**    To more intuitively demonstrate AWM's cross-template generalization and ability to build increasingly complex workflows (as exemplified in Figure 1), we conduct a case study to illustrate the workflow mechanism behind it.

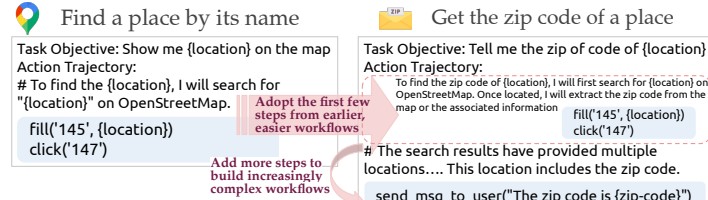

Figure 6: AWM builds increasingly complex workflows over time, by learning from past examples and earlier workflows.

As exemplified in Figure 6, the agent creates and learns the "Find a place by its name" workflow in the early stage of the online process by summarizing past examples. Later, when encountering an example that further asks to obtain the zip code of the location, AWM agent learns to adopt the first few steps to find locations by following the existing workflow, and then conducts further steps to obtain the zip code of the place found. Integrating these new steps upon the vanilla find location task, the agent successfully builds a more complex workflow, i.e., "get the zip code of a place".

## 3.2 MIND2WEB

Mind2Web (Deng et al., 2023) features web navigation in cross-task, website, and domain settings, stressing the generality of agents on versatile operations and environments. Each task in Mind2Web has a fixed number of steps; at each step, the agent needs to predict an action, which is evaluated by: (1) *element accuracy*: to check if the correct page element is selected, (2) *action $F_1$* to check if the action taken on the element is correct, and aggregating (1) and (2) yields (3) *step success rate* which checks that both element and action selection are correct at the current step. Lastly, after completing every step in the given task, the last metric (4) task-level *success rate* measures if all intermediate steps are successfully conducted for this task, i.e., all steps for this task score 1 under metric (3).

Because Mind2Web provides a training set covering part of the tested websites (cross-task split), we explore both the *offline* setting that induces workflows from the training set and applies to test sets, and the *online* setting, where we stream workflow induction and inference on test queries (§2.3).

Since we focus on LM-based agents that only take textual inputs, we compare AWM to two state-of-the-art methods, MindAct (Deng et al., 2023) and Synapse (Zheng et al., 2024). MindAct introduces webpage element filtering and multi-choice task format to ease observation processing, and Synapse changes the format to a trajectory style and augments retrieved relevant examples. We integrate the element filtering adopted in both methods, and added workflows instead of retrieved examples in Synapse, to verify the superiority of reusable workflows over concrete examples. To fairly compare with all baseline methods, we run AWM with both `gpt-3.5-turbo` and `gpt-4` models with temperature 0.0. We use the same model for neural workflow induction and agent action generation.

Table 4: Success rate on Mind2Web cross-task, cross-website, and cross-domain generalization test, using `gpt-4` model. EA is short for element accuracy and $AF_1$ is short for action $F_1$.

| Method | Cross-Task | | | | Cross-Website | | | | Cross-Domain | | | |
|---|---|---|---|---|---|---|---|---|---|---|---|---|
| | EA | $AF_1$ | Step SR | SR | EA | $AF_1$ | Step SR | SR | EA | $AF_1$ | Step SR | SR |
| MindAct* | 41.6 | **60.6** | 36.2 | 2.0 | 35.8 | **51.1** | 30.1 | 2.0 | 21.6 | **52.8** | 18.6 | 1.0 |
| $AWM_{offline}$ | **50.6** | 57.3 | **45.1** | **4.8** | 41.4 | 46.2 | 33.7 | **2.3** | 36.4 | 41.6 | 32.6 | 0.7 |
| $AWM_{online}$ | 50.0 | 56.4 | 43.6 | 4.0 | **42.1** | 45.1 | **33.9** | 1.6 | **40.9** | 46.3 | **35.5** | **1.7** |

### 3.2.1 MAIN RESULTS

We first run with AWM offline using both GPT variants, and find that AWM consistently obtains the highest success rate in both step and task levels, leading to $4.0$–$8.9\%$ relative and $0.4$–$2.8$ absolute points increases in step-wise and task-wise success rates than the baselines — Synapse with `gpt-3.5-turbo` and MindAct with `gpt-4`. Decomposing the step success rate by element and action selection and accuracy, we notice the increases mainly come from more accurate element selection, as indicated by the $5.0$–$9.0$ element accuracy increase in Table 3.

Table 3: AWM offline results on Mind2Web cross-task dataset. Elem Acc and SR are short for element accuracy and success rate. We footnote the GPT variant used by each method, `3.5` and `4` stands for `gpt-3.5-turbo` and `gpt-4`, respectively. We highlight the best result within the same model variant.

| Method | Elem Acc | Action $F_1$ | Step SR | SR |
|---|---|---|---|---|
| $MindAct_{3.5}$ | 20.3 | **56.6** | 17.4 | 0.8 |
| $CogAgent_{3.5}$ | - | - | 18.6 | - |
| $Synapse_{3.5}$ | 34.0 | - | 30.6 | 2.4 |
| $AWM_{3.5}$ | **39.0** | 52.8 | **34.6** | **2.8** |
| $MindAct_4$ | 41.6 | **60.6** | 36.2 | 2.0 |
| $AWM_4$ | **50.6** | 57.3 | **45.1** | **4.8** |

**Abstract sub-routines vs. concrete experiences** More specifically, compared to the Synapse (Zheng et al., 2024) method that retrieves the most relevant training examples, AWM achieves a $+5.0$ element accuracy and leads to a $+4.0$ increase in step success rate. While augmenting concrete, full examples may bias agents to select elements similar to those presented in the given examples, AWM introduces less bias on element selection via its abstract representation of example-specific contexts in workflows, and therefore enables higher step success rates.

Furthermore, AWM integrates frequently-used sub-routines, which can be more flexibly and readily leveraged across test examples, compared to the full example trajectories used by Synapse, which are less likely to appear multiple times. In general, our results indicate that the abstract, reusable nature of workflows contributes to the superiority of the AWM method.

**Learn to diverge from workflow guidelines** Despite more accurate element selection, AWM gets slightly lower action $F_1$ scores than MindAct, possibly because the augmented workflows may guide the agent to take certain actions aligning to the workflows, which are not always relevant to the particular environment state at hand. While following the workflows generally results in more successful task trajectories, agents still encounter some challenges in identifying places to diverge from the workflow guidelines.

### 3.2.2 ONLINE AWM ENABLES GENERALIZATION

Beyond the offline induction setting, we further explore the AWM in the online setting, similar to the WebArena experiment setup in §3.1, where no additional training examples are needed besides test queries. This naturally facilitates cross-website and cross-domain generalization, which we examine on the two other splits provided by the Mind2Web dataset: cross-website and cross-domain tests.

In addition to the MindAct baseline, we additionally set bars with our $AWM_{offline}$ setup, by randomly selecting workflows induced from the training set as memory augmentations. Specifically, for cross-website examples, we select workflows from the same domain; for the cross-domain setting, we randomly select workflows from all domains. We conduct $AWM_{online}$ by iteratively inducing, integrating, and utilizing workflows over test inferences. We also explore $AWM_{offline+online}$ in §C.

As shown in Table 4, both $AWM_{online}$ and $AWM_{offline}$ surpass the MindAct baseline by a large margin, resulting in $7.4$–$8.9$, $3.6$–$3.8$, and $14.0$–$16.9$ absolute point improvements in step success rates, in cross-task, cross-website, and cross-domain scenarios.

**In-domain, cross-task scenario** When tested in-domain, $AWM_{online}$ and $AWM_{offline}$ perform comparably to each other. When inspecting the model behaviors in detail, we notice the pros and cons of each method. $AWM_{online}$ induces workflows from model-predicted trajectories that are not always correct, thus can lead to incorrect workflows that degrade model performance. On the other hand, the training and test examples on some websites vary in task distributions (e.g., training examples cover how to buy items on Amazon, test examples ask for job applications to Amazon careers.). $AWM_{online}$ naturally resolves this train-test gap because its operating process only involves test queries and environments, therefore yields workflows that are presumably more targeted toward the test distribution, which in turn, leads to higher success rates overall. Nonetheless, if distribution-matching, high-quality training examples are available, $AWM_{offline}$ could bring more benefit by alleviating the gap issue, as the slightly higher cross-tasks scores of $AWM_{offline}$ in Table 4.

**Extending to unseen websites and domains** When applied on unseen websites or domains, $AWM_{online}$ demonstrates greater generalization abilities, compared to $AWM_{offline}$. The performance margin of $AWM_{online}$ (over $AWM_{offline}$) widens as the domain gaps between training and testing data widen from different websites (e.g., *apple* to *bestbuy*) to different domains (e.g., *macys* in shopping domain to *reddit* in social media domain). Because $AWM_{online}$ does not require nor rely on information from the training data, it is not affected by any domain gaps. Nonetheless, as demonstrated by the substantial improvements of $AWM_{offline}$ over the MindAct baseline, $AWM_{offline}$ still demonstrates that models can benefit from mechanistically similar workflows from the previously induced workflow repository.

## 4 EXPLORING OPTIMAL WORKFLOW REPRESENTATIONS

In this section, we experiment with other possible alternatives to better represent the workflows. Specifically, we ablate workflows in sub-routine, abstract formats (§4.1), explore workflows in descriptive texts (§4.2), and lastly, beyond the default workflows that describe environment state in NL, we compare strengthened observations with website HTML within workflow steps (§4.3).

### 4.1 HOW MUCH DOES THE SUB-ROUTINE, ABSTRACT FORMAT CONTRIBUTE?

In this section, we compare our abstract, sub-routine-based induction method using LMs to a rule-based method without context and sub-routine abstraction.

Specifically, our rule-based induction $I_{rule}$ first extracts the action sequence (e.g., CLICK $\rightarrow$ CLICK $\rightarrow$ TYPE) of each experience and deduplicates experiences by their action sequence. In each unique experience, we then remove the steps whose action cannot be executed on the environment. We take these unique, validated experiences as workflows. Find more detailed descriptions in §B.

**WebArena Results** As shown in Table 5, using rule- and LM-based workflow induction performs comparably, with a small $0.1$ gap in success rate; the LM-based method appears more efficient and uses $0.4$ fewer steps. Our manual analysis found workflows produced by the LM-based induction module $I_{lm}$ are finer-grained, preventing agents from following unnecessary steps that sometimes appear in rule-induced workflows, hence making the task-solving process slightly more efficient.

Table 5: AWM success rate on WebArena using gpt-4, with rule- and lm-based induction.

| Method | Total SR | # Steps |
|---|---|---|
| $AWM_{rule}$ | **35.6** | 6.3 |
| $AWM_{lm}$ | 35.5 | **5.9** |

**Mind2Web Results** In Table 6, compared to $AWM_{rule}$, $AWM_{lm}$ improves by a 2.8 margin. While augmenting concrete, full examples may bias agents to select elements similar to those presented in the given examples, $AWM_{lm}$ introduces less bias on element selection via its abstract representation of example-specific contexts in workflows.

Table 6: AWM results with different workflow induction methods on Mind2Web cross-task dataset.

| Method | Elem Acc | Action $F_1$ | Step SR | SR |
|---|---|---|---|---|
| $MindAct_4$ | 41.6 | **60.6** | 36.2 | 2.0 |
| $AWM_{4,rule}$ | 49.5 | 57.0 | 43.4 | 2.0 |
| $AWM_{4,lm}$ | **50.6** | 57.3 | **45.1** | **4.8** |

Further, $AWM_{lm}$ uses frequently-used sub-routines, which can be more flexibly and readily utilized across test examples, compared to the full example trajectories induced by $AWM_{rule}$, which are less likely to appear multiple times. In general, our results indicate that the abstract, reusable nature of workflows contributes to the efficacy of $AWM_{lm}$ method.

## 4.2 Workflows in Descriptive Texts

AWM represents workflow steps in a program format. In this section, we compare with a textual format for workflows, to understand whether text or code serves as a better format for agent memory. More concretely, we prompt `gpt-3.5-turbo` to verbalize the action trajectory in the workflows induced in earlier experiments. For example, from an action `CLICK({submit-id})`, its verbalized NL representation reads similar to "`CLICK` the submit button". We use the same textual observation and thoughts from code actions as observation and thoughts in these text actions.

From the results in Table 7, $AWM_{text}$ achieves slightly higher element selection accuracy and step success rate, by 0.6 and 0.3 points, respectively, yet degrades 1.2 in task success rate. Overall, we do not find substantial performance variance between workflows represented in text and code formats, indicating that both forms can bring effective augmentations to agent memory.

Table 7: Mind2Web cross-task results with AWM using code and text workflows.

| Method | Elem Acc | Action $F_1$ | Step SR | SR |
|---|---|---|---|---|
| MindAct | 41.6 | **60.6** | 36.2 | 2.0 |
| AWM | 50.6 | 57.3 | 45.1 | **4.8** |
| $AWM_{text}$ | **51.2** | 57.4 | **45.4** | 3.6 |

## 4.3 Environment Abstraction in Workflows

AWM describes intermediate webpage states using NL, yet showing concrete states may be helpful to better ground agents on the environment. Since a webpage's full HTML can be overly long, we filter the webpage representation using the relevance predictor of Deng et al. (2023), and augment each workflow step with this shortened HTML that only has elements predicted as relevant. We run `gpt-3.5-turbo` with only descriptions, only HTML, and both types of content.

As shown in Table 8, NL description of states is more useful than HTML, as replacing NL with HTML leads to a slight 0.8 drop in step success rate. Interestingly, using both NL and filtered HTML leads to worse results. We conjecture the reason to be two-fold. First, adding NL and HTML substantially in-

Table 8: Mind2Web results using GPT-3.5-turbo with different environment representations.

| Desc. | HTML | Elem Acc | Act $F_1$ | Step SR | SR |
|---|---|---|---|---|---|
| ✓ | ✗ | **39.0** | 52.8 | **34.6** | **2.8** |
| ✗ | ✓ | 38.1 | **54.0** | 33.8 | **2.8** |
| ✓ | ✓ | 37.1 | 51.3 | 32.9 | 2.0 |

creases the context length, thus making it harder for models to handle things correctly. Second, the filtered HTML has a substantial number of irrelevant items (missing all correct elements 47% of the time) thus potentially contradicting NL descriptions and impairing agent abilities.

## 5 Exploring Workflow Utilization in Context and in Action

Besides integrating workflows as agent memory, we also explore workflows in expanding the agent action space, denoted as $AWM_{AS}$. We leverage the programmatic nature of workflows and wrap each workflow into a high-level function, similar to a shortcut tool the agent can call to perform a pre-determined series of actions (Wang et al., 2024b). Formally, an agent is initially equipped with default, primitive actions $P$ (e.g., `click`, `type`), and $AWM_{AS}$ adds the induced workflow actions $W$ (e.g., `find_place`, `get_place_zipcode`) to its action space.

The agent can call a primitive or workflow action at each step. When a primitive action is called, the agent immediately takes that action. When the agent calls a workflow action, it will trigger the sequence of pre-determined steps in the workflow. For example, calling the `login(username, password)` workflow action results

Table 9: Mind2Web results with $AWM_{AS}$ variant that alters the action space besides memory augmentation. All methods use `gpt-4`.

| Method | Elem Acc | Action $F_1$ | Step SR | SR |
|---|---|---|---|---|
| MindAct | 41.6 | **60.6** | 36.2 | 2.0 |
| AWM | 50.6 | 57.3 | 45.1 | **4.8** |
| $AWM_{AS}$ | **51.8** | 56.7 | **46.4** | 3.6 |

in sequentially executing `click(box1-id)` → `type(box1-id, username)` → `click(box2-id)` → `type(box2-id, password)` → `click(submit-id)`. The workflow action is completed when all intermediate primitive actions are finished.

In Table 9, expanding the agent action space with workflows ($AWM_{AS}$) slightly improves the step success rate by 1.3 points, and gets the same overall success rate, 3.2, of the base memory-

augmented AWM. We analyzed agent predictions and found they call workflow actions in merely 18.5% of the tasks, suggesting a resistance of current agents to use newly-added actions. Overall, expanding actions with workflows seems to reinforce workflows in memory, and brings small extra gains as auxiliary actions.

However, workflow actions do not always lead to task success. A representative example is shown in Figure 7. When booking flights, users often input a city name such as "New York," yet the system often pops up some nearby airports to support next-step search. While one can induce a `book_flight` workflow that enters all required data via a pre-determined action sequence, the action to

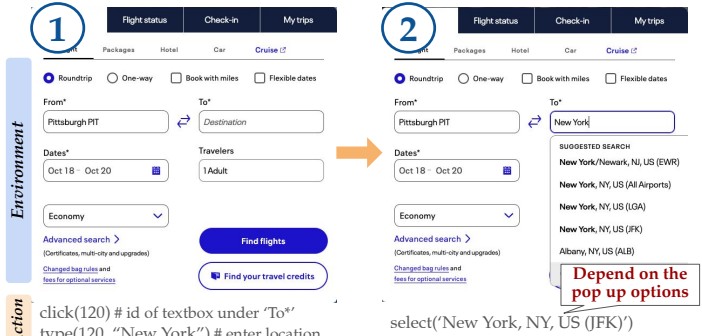

Figure 7: An example of dynamic environment changes that challenge workflow action utilization.

choose pop-up airports is executed without seeing the intermediate states with available pop-up options, and is not flexible enough to do so. More advanced techniques such as granting real-time state access or dynamic execution loops can be promising to solve this issue, and we encourage future work to leverage the AWM framework to explore these.

## 6 RELATED WORK

**Web Agent Benchmarks** The first modern and widely used web agent benchmark is Shi et al. (2017)'s MiniWob, which evaluates across various scenarios such as flight booking. (Liu et al., 2018) then created MiniWob++ with extra challenges. More recently, WebShop (Yao et al., 2022) features a simulated e-commerce website and crowd-sourced text instructions. WebArena (Zhou et al., 2024) integrates four more websites and enables realistic execution-based evaluations, and VisualWebArena (Koh et al., 2024) extends with tasks that necessitate visual inputs. Mind2Web (Deng et al., 2023) proposes versatile tasks and stresses agent generalization across websites and domains. We use WebArena and Mind2Web to evaluate our method's task success and generality.

**Enhancing Agents for Complex Tasks** Many works improve agents by modifying their action space, such as constraining its action search space (Liu et al., 2018), enabling LM self-feedback to refine predicted actions (Sun et al., 2023), or incorporating human-designed actions to certain tasks (Sodhi et al., 2023; Sarch et al., 2024). Other works explore ways to augment agent memory, such as adding example demonstrations in context (Haluptzok et al., 2023; Zheng et al., 2024; Fu et al., 2024). However, high-quality examples are not always available or easy to collect. Our AWM can flexibly operate even when auxiliary examples are non-existent and only test queries are available.

**Learning Common Procedures from Experiences** Some works use full examples (Zheng et al., 2024) as context for an agent, yet they entangle with example-specific contexts and face challenges in extrapolating to other tasks or domains (Majumder et al., 2023). Many works propose to extract frequently reused sub-routines from experiences with rule-based (Ellis et al., 2023; Bowers et al., 2023; Grand et al., 2023) or LM-based methods (Cai et al., 2023; Wang et al., 2024c;a) methods, and use them as auxiliary skills to ease future task-solving (Oh et al., 2017; Liang et al., 2023; Yu et al., 2023; Mao et al., 2023). We explored both rule- and LM-based methods to induce reusable workflows, and use them flexibly as context guidance that are free of environment grounding issues.

## 7 CONCLUSION

We propose agent workflow memory that induces, augments, and uses workflows, offline from available examples or purely online at inference time. We evaluate AWM on WebArena and Mind2Web, and achieve 24.6% and 51.1% relative increases in task success rate. AWM also demonstrates its superior generalization abilities across tasks, websites, and domains. We hope AWM sheds insight on and boosts advances in dynamic memory building and agent adaptations on varied digital tasks.

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

# A  LM-BASED WORKFLOW INDUCTION

As introduced in §2.3, one realization of our workflow induction module is to prompt LMs to generate abstract, sub-routine workflows from the given examples, i.e., experience. In this section, we provide the detailed model prompt, exemplar workflows induced by models, and quality examination on these workflows.

## A.1  MODEL PROMPT

We provide the exact prompt inputted to the model for WebArena and Mind2Web experiments below. Experiments on both datasets use the same prompt.

> Given a list of web navigation tasks, your task is to extract the common workflows.
> Each given task contains a natural language instruction, and a series of actions to solve the task. You need to find the repetitive subset of actions across multiple tasks, and extract each of them out as a workflow.
> Each workflow should be a commonly reused sub-routine of the tasks. Do not generate similar or overlapping workflows. Each workflow should have at least two steps. Represent the non-fixed elements (input text, button strings) with descriptive variable names as shown in the example.

## A.2  EXAMPLE WORKFLOWS

We present several exemplar workflows induced on WebArena and Mind2Web, to give a more concrete impression of workflows.

**WebArena Workflows**    We show one example workflow on each website involved in WebArena.

> ## shopping: Browse Products in a Specific Category
>
> To browse products in a specific category, I need to navigate to the relevant main category. I will start by hovering over the main category menu item to reveal the subcategories.
> ```
> hover('main_category_id')
> ```
> To browse products in the specific subcategory, I need to click on the subcategory link.
> ```
> click('subcategory_id')
> ```

> ## shopping admin: Edit and Save Changes
> This workflow is used to edit specific fields and save changes.
>
> To edit a specific field, I need to locate the field and update its value.
> ```
> clear('field_id')
> fill('field_id', 'new_value')
> ```
> Next, I need to save the changes by clicking the "Save" button.
> ```
> click('save_button_id')
> ```

> ## reddit: Navigate to a forum section and select a specific forum
>
> To navigate to a specific forum, I need to click on the "Forums" section.
> ```
> click('42')
> ```
> Now, I need to click on the specific forum link based on the forum name provided.
> ```
> click('<forum_link_id>')
> ```

> ## gitlab: Navigation to Repository and Contributors Section
> This workflow involves searching for a repository and navigating to its contributors to find detailed contribution data.
>
> First, search for the specific repository to gather information.
> ```
> fill('130', '{RepositoryName}')
> press('130', 'Enter')
> ```
> Navigate to the "Contributors" section to view contribution details.
> ```
> click('311')   # "Contributors" link
> ```
> Obtain and report the required contributor details.
> ```
> send_msg_to_user('{ContributorDetails}')
> ```

```
## map: Calculate Travel Time and Distance
To calculate travel time and distance between two locations, I will use the directions feature. I
will fill in the respective fields and select the mode of transportation.
fill('158', 'FROM_LOCATION')
fill('163', 'TO_LOCATION')
select_option('166', 'MODE_OF_TRANSPORTATION')
click('171')
I will use these details to provide the user with accurate travel time and distance information.
send_msg_to_user('The distance between FROM_LOCATION and
TO_LOCATION is DISTANCE and the estimated travel time is TIME.')
```

**Mind2Web Workflows**  We present one example workflow in each data domain in Mind2Web.

```
# travel: enter_flight_locations
Given that you are on the flight booking page, this workflow enters the departure and destination
city/airport for your flight.
[link] From Departure Airport or City Your Origin − > CLICK
[textbox] Origin City or Airport − > TYPE: {your-origin-city}
[link] {best-popup-option} − > CLICK
[link] To Destination Airport or City Your Destination − > CLICK
[textbox] Destination City or Airport − > TYPE: {your-destination-city}
[link] {best-popup-option} − > CLICK
```

```
# shopping: search_and_sort
Given that you are on the Amazon search results page, this workflow searches for a product and
sorts the results.
[textbox] Search Amazon − > TYPE: {search-term}
[button] Go − > CLICK
[span] Sort by: − > CLICK
[option] {sort-option} − > CLICK
```

```
# entertainment: search_and_select
Given that you are on the IMDb homepage, this workflow searches for a term and selects the best
match.
[textbox] Search IMDb − > TYPE: {search-term}
[button] Submit Search − > CLICK
[button] {best-match} − > CLICK
```

### A.3 WORKFLOW QUALITY ANALYSIS

To provide intermediate information beyond the end-to-end task success, we propose several metrics to verify the quality of the model-induced workflows. (1) *Number of workflows*: The number of workflows augmented to the memory, fewer workflows is better, whereas agents rely on fewer workflows to achieve satisfactory performance. (2) *Coverage*: How many steps in the action trajectory are covered by the workflows, higher coverage presumably signals the general applicability of the concerned workflow. (3) *Function overlap*: How much functionality overlap exists between workflows, we measure this by counting the number of overlapping sub-trajectories ($\leq 2$ steps) between each workflow pair for the same website. Less overlap indicates more maximized workflow management. (4) *Utility rate*: How often are workflows used by test examples.

We evaluate the workflows on WebArena test examples and Mind2Web cross-task test examples. We do not evaluate coverage on WebArena since it requires canonical trajectories, yet which are not available for WebArena. For Mind2Web, we do not evaluate on cross-website and cross-domain test examples since workflows induced from training examples do not have domain overlapping with these test examples, thus less applicable to them.

As shown in Table 10, neural-based induction produces 7.3–7.4 workflows per example, which is efficient and do not add too much content to the memory. On WebArena, the induced workflows are

Table 10: Quality evaluation of model-induced workflows on Mind2Web dataset.

| Metric | # Workflows | Coverage | Function Overlap | Utility Rate |
|--------|-------------|----------|------------------|--------------|
| WebArena | 7.4 | - | 0.08 | 0.94 |
| Mind2Web | 7.3 | 0.40 | 0.20 | 0.91 |

used by 0.94 of the test examples, indicating its wide applicability among varied tasks. Further, only 0.08 of the steps between workflows overlap, demonstrating the efficiency of workflows in solving respective tasks. Workflows on Mind2Web, although used similarly frequently as indicated by the high 0.91 utility rate, have slightly more functional overlap, and only achieve a 0.40 coverage over test examples. However, as the training examples used to induce workflows have substantial task distribution variances with the cross-task test examples, this relatively low coverage is reasonable.

## B  RULE-BASED WORKFLOW INDUCTION

Beyond LM-based workflow induction, we also explored a rule-based workflow induction method. Our rule-based workflow induction module consists of two steps: (i) experience deduplication, and (2) invalid action filtering.

For deduplication, we extract the action sequence of the experience, e.g., extracting CLICK → CLICK → TYPE from the trajectory CLICK('12') → CLICK('30') → TYPE('44', "cat"). We group experiences by their action sequence and randomly select $n$ ($n$ = 1 by default) experiences from each group. Specifically on WebArena, where the task template for each experience is available. We conduct another round of deduplication by grouping experiences by their task template, and randomly selecting $n$ ($n$ = 1 by default) experiences from each group. This process yields diverse experiences from the given set of experiences.

Next, for each unique experience, we remove the invalid steps in its action trajectory. Invalid actions means actions that cannot be successfully executed on the environment, because the input arguments do not meet the requirement of the action function. Specifically, we have one rule of determining invalid actions for CLICK and TYPE, that requires the first argument to be a string-formatted integer (which refers to the id of an element in the environment). We remove CLICK and TYPE steps if they do not meet this requirement. For example, an experience with trajectory CLICK(12) → CLICK('12') → CLICK('30') → TYPE(44, "cat") → TYPE('44', "cat") will yield CLICK('12') → CLICK('30') → TYPE('44', "cat"). We conduct this invalid action filtering for each unique experience, and take the resulting experiences as rule-based workflows.

## C  INTEGRATING AWM OFFLINE AND ONLINE

We compared AWM$_{offline}$ and AWM$_{online}$ in §3.2, that adopts workflows induced separately from training or on-the-fly during testing, respectively. In this section, we explore an integration of both sets of workflows, AWM$_{off+on}$, that injects relevant training workflows to warm start task-solving, but also aggregates increasingly more online-induced workflows to better adapt to test distributions.

Table 11: Success rate on Mind2Web cross-task, cross-website, and cross-domain generalization test, using gpt-4 model. EA is short for element accuracy and AF$_1$ is short for action F$_1$.

| Method | Cross-Task | | | | Cross-Website | | | | Cross-Domain | | | |
|--------|-----|-----|---------|-----|-----|-----|---------|-----|-----|-----|---------|-----|
| | EA | AF$_1$ | Step SR | SR | EA | AF$_1$ | Step SR | SR | EA | AF$_1$ | Step SR | SR |
| MindAct* | 41.6 | **60.6** | 36.2 | 2.0 | 35.8 | **51.1** | 30.1 | 2.0 | 21.6 | **52.8** | 18.6 | 1.0 |
| AWM$_{offline}$ | **50.6** | 57.3 | **45.1** | **4.8** | 41.4 | 46.2 | 33.7 | **2.3** | 36.4 | 41.6 | 32.6 | 0.7 |
| AWM$_{online}$ | 50.0 | 56.4 | 43.6 | 4.0 | **42.1** | 45.1 | **33.9** | 1.6 | **40.9** | 46.3 | **35.5** | **1.7** |
| AWM$_{off+on}$ | 50.0 | 57.0 | 44.5 | 1.6 | 41.8 | 45.5 | 33.3 | 1.1 | 39.3 | 44.3 | 34.1 | 1.5 |

From Table 11, AWM$_{off+on}$ scores between AWM$_{offline}$ and AWM$_{online}$ across three test splits. Rather than an additive effect, workflows induced offline and online are not fully compatible with each other, particularly, the offline workflows seem to impair the generative quality and utility efficacy of online workflows, therefore resulting in medium results overall.

