# OpenReview forum: "Agent Workflow Memory"
_ICLR.cc/2025/Conference — Submitted to ICLR 2025_

### Official Review · Reviewer_fnjL · 2024-11-04

**Soundness:** 2
**Presentation:** 2
**Contribution:** 2
**Rating:** 5
**Confidence:** 4

**Summary:**

The authors propose an agent workflow memory that induces, augments and uses the most common functions as part of an existing library that the agent can take advantage of. This can be operated in both offline and online scenarios i.e. in the presence or absence of labeled samples. They evaluate the approach on two challenging  web navigation benchmarks mind2web and web arena and show that this simple and effective approach generalizes well across web navigation benchmarks.

**Strengths:**

#### Strengths:
1. **IMemory Utilization:**
   - AWM's ability to integrate workflows into the agent’s memory, allowing for more sophisticated task handling and adaptation over time, is noteworthy. This is an advancement in making agents more autonomous and capable

2. **Empirical Validation:**
   - The empirical results showing AWM's performance boosts of 24.6% and 51.1% in relative success rates on the Mind2Web and WebArena benchmarks respectively are impressive. These results provide evidence of the workflow memory working to some level of competency.

3. **Generalization across Domains:**
   - The method's robustness across tasks, websites, and domains, particularly in environments where no annotated examples exist, highlights its potential for real-world applications and its adaptability.

**Weaknesses:**

#### Areas for Improvement, please take this with a pinch of salt:
1. **Writing and Clarity:**
 - the paper needs a good amount of work on writing to meet the publication standards.  Can avoid unnecessary floating images and tables. Perhaps can save space by removing much of the unnecessary text and allowing natural space. More importantly, the overall story and clarity and motivation at places need improvement on why something is being discussed.

2. **Workflow Induction Clarity:**
   - The work could benefit from a more detailed explanation of the workflow induction process, particularly how the workflows are selected and refined during the induction phase. More insights into the criteria for choosing particular workflows over others would enhance the reader's understanding of the method's inner workings.

3. **Discussion on Limitations:**
   - While the paper discusses the success of the AWM model, a more thorough exploration of its limitations, especially in scenarios where the agent encounters completely novel tasks or domains not covered by the workflows, would provide a balanced view of its applicability.

4. **Comparison with State-of-the-Art Methods:**
   - Expanding the comparison with current state-of-the-art methods, not just in terms of task success rates but also considering factors like computational efficiency and scalability, would provide a clearer picture of standings.

**Questions:**

In the feedback from fellow reviewers and the authors, I would appreciate a discussion or clarity on the terminology used in the paper. The approach described appears to be a knowledge-retaining technique, akin to the reuse of skills stored in a skills library for related tasks, as seen in other literature such as Voyager (Wang et al., 2023) or works on counterfactual reasoning. These sources typically refer to reused routines as "skills" rather than "workflows," and use "library" instead of "memory" for storage. It would be beneficial for the paper to acknowledge its inspirations from prior works, especially in the context of web navigation applications. This acknowledgment could help clarify the rationale behind the choice of terminology right from the start.

---

> ### Author Response · Authors · 2024-11-17
>
> Thank you for your recognition in the novelty of our AWM method design, the effectiveness shown by extensive experiments, and AWM’s strong generalization abilities.
>
> **W1: Writing needs to be improved**
> We appreciate your comments. However, since the feedback is somewhat general, and given that other reviewers have noted the clarity of our writing, we would greatly appreciate any additional details you could provide. If you could clarify specific aspects of the story or identify areas where motivation and clarity could be further strengthened, we would be more than happy to make those improvements.
>
> **W2: Workflow Induction Clarity**
> The workflow induction process prompts an LLM with examples ((NL, trajectory)) pairs, and expects it to output workflows (as defined in section 2.2). Please find more discussions in formalized ways (throughout section 2.3), in graphical ways (Figure 2, 3, and 4), in natural language descriptions (throughout section 2.3), and with concrete examples (Appendix A). Please let us know which dimension you found unclear so that we can improve on it!
> Also, we would like to clarify that we **do not** “choose particular workflows over the others” (as stated by the reviewer). Our method incorporates all induced workflows into agent memory without explicit selection.
>
> **W3: Discuss limitations**
> We will discuss more limitations of our AWM approach, including its current limited capability in generalizing across tasks/domains,
> **W4: Considering factors like efficiency and scalability**
> Thank you for recognizing the importance of efficiency/scalability beyond task success rate. We agree with this and **have already evaluated** this dimension, by reporting the number of steps the agent took to solve the task, in the “# steps” column in Table 1 — which shows AWM takes fewer steps thus has better efficiency.
> Besides the action generation step, our AWM approach adds two other steps – trajectory evaluation and workflow induction. We calculate the average compute of all three steps by the number of input/output/total tokens per step, the average number of times that the step occurs per task, and the total number of tokens used on average for a task.
> As shown in the table below, the trajectory evaluation step and workflow induction step only take 4.0% and 6.8% of the compute of the original action generation step. Compared to the baseline method (using action generation step only), our AWM approach only adds 10.8% computation overhead, but brings a 51.5% accuracy increase in WebArena tasks, demonstrating the cost-effectiveness of our AWM approach.
>
> | Step | # Input Tokens Per Step | # Output Tokens Per Step | # Tokens Per Step | # Occurrence | # Total Tokens |
> | - | - | - | - | - | - |
> | Action Generation | 5,663 | 52 | 5,715 | 5.9 | 33,718.5 |
> | Trajectory Evaluation | 306.8 | 82.8 | 389.6 | 5.9 | 2,298.6 |
> | Workflow Induction | 306.8 | 328.7 | 635.5 | 2.1 | 1344.6 |
>
> Furthermore, we carefully study the learning process of our AWM agent and plot its learning curve after processing an increasing number of examples, in Figure 5 and Figure 1. We show that our AWM approach enables agents to quickly acquire essential workflows, thus superior skills in task-solving, by processing **only tens of examples (<40)** with **zero supervision**, demonstrating the scalability of our method.
>
>
> **Q1: Discussion of the terminology used**
> We agree and **have acknowledged** these relevant inspiring works in a full paragraph (line 524–532) in section 6.
> Nonetheless, we would like to clarify a key difference in our “agent memory”, and why we favor it against the “skills” and “library” in previous works such as Voyager: our “agent memory” presents in text, which provides several benefits that programmatic “skills” do not have: (1) less requirement in accuracy: the text can be free-form and verbalized in flexible ways, without having to be executed successfully, alleviating the requirement on induction step; (2) our workflows can easily generalize across websites and domains, whereas programmatic skills need to be modified to execute on other websites.
> Furthermore, we have demonstrated turning workflows into actions, i.e., “skills”, in Section 5, and showing its limited applicability in web navigation tasks, as of now.
>
> In short, we do appreciate and acknowledge these inspiring works, but proposed our textual workflow memory that can solve the web navigation task more flexibly and accurately.

---

> > ### Comment · Reviewer_fnjL · 2024-11-25
> >
> > This reviewer appreciates the response and added explanations where necessary. However, reservations remain about the thoroughness of the work and minor concerns regarding the paper's publication readiness, as I mentioned earlier about floating figures 1, 2, 3, 4, and 6 and unnecessary crammed text. I am ready to incline and discuss with other reviewers and senior chairs if they agree that the contributions are noteworthy and do not warrant a fresh round of reviews for improvement.

---

### Official Review · Reviewer_Wfe1 · 2024-11-05

**Soundness:** 2
**Presentation:** 3
**Contribution:** 2
**Rating:** 5
**Confidence:** 4

**Summary:**

This work proposes agent workflow memory as an approach to building web agents that accumulate knowledge from their experiences as workflows. Specifically, it converts each successful trajectory it obtains to a workflow, which consists of a workflow description and workflow trajectory. It stores those workflows on a per-website basis. For inference, the corresponding workflows are incorporated into the agent's input and used for solving the tasks. The authors conduct experiments on WebArena and Mind2Web and demonstrate that their approach outperforms the compared baseline methods.

**Strengths:**

- Acquiring domain-specific knowledge in a re-usable form can be useful and the proposed approach is reasonable. Especially, the authors try to acquire shorter chunks of re-usable skills as workflows, which could be important for inceasing the utility of such workflows.
- Overall, the writing is easy to follow, especially with the well-visualized figures.
- They perform additional empirical analyses, such as the form of workflows and how useful those acquired workflows are. These kinds of analyses can be helpful in constructing a deeper understanding of the proposed approach.

**Weaknesses:**

- The proposed approach accumulates experiences as workflows on a website basis (L196). While this can be effective in focusing on a specific set of websites, it also comes with a limitation in terms of generalization to different websites. This may also become a practical limitation, as there are myriad websites available and preparing various scenarios for most of them could be quite challenging. On benchmarks, this can be a factor that makes the empirical evaluation of the generalization capabilities of this approach difficult. Although this work tests the proposed approach in the online setting, existing benchmarks are based on pre-defined sets of websites.
- Although the online acquisition of workflows in a continual manner can be helpful in practice, comparing only the online version with the baseline approaches on the benchmark tasks can be unfair or misleading. Usually, getting information about other test tasks can bring noticeable improvement in the performance in most benchmarks, mainly because of the similarities in the tasks and the bigger data size. For a fairer comparison without the knowledge about the "held-out" test tasks, one can try gathering experiences (and thus workflows) without access to any of the tasks from the test set and evaluating on each of the test tasks without updating the set of workflows.
- In Table 4 for the Mind2Web results, the numbers for MindAct in the cross-domain split are the GPT-3.5 results, not the GPT-4 results.

**Questions:**

Please take a look at the weaknesses section.

---

> ### Author Response · Authors · 2024-11-17
>
> Thank you for recognizing our reasonable method design, clear writing and figures, as well as the extensive empirical analysis throughout the sections.
>
> **W1: Limitation in generalizing to different websites**
> First of all, we do see improvements in generalization with our AWM approach in (i) cross-website transfer in WebArena (Section 3.1.3), and generalization across tasks, websites, and domains on Mind2Web (Section 3.2.2), compared to all baseline methods.
> We design AWM exactly because of the limited generalization of LLMs out-of-the-box, by enabling agents to learn new workflows starting with zero experience on an unseen website.
>
>
> **W2: Unfair online setting**
> We argue that **our online setting is fair to compare** with baseline methods. AWM has the same information as the baseline methods (i.e., all test queries), only has access to test queries, and no annotated solutions for any problems are needed. Our method simply changes the way the agent processes the test queries — by deriving and reusing workflows on the fly, instead of processing each test query independently.
> More importantly, we believe **our online setting is a more realistic setting for evaluating agents**, where user queries come in a stream, and no/little supervision is provided.
> In many other reasoning/agent tasks, people are now beginning to explore other ways to synergize multiple test examples rather than treating them independently, such as batched prompting [1]. We think these different ways (including ours) to process test queries are valid and constitute fair comparisons.
>
> Despite the online setting, our offline AWM approach also achieves superior performance in task success and generalization, showing the effectiveness of our method.
>
> **W3: Table 4 MindAct cross-domain results reports gpt-3.5**
> Thank you so much for pointing this out! We will update the results to GPT-4 scores.
> In particular, the GPT-4 step success rate is 26.4, which is still lower than our AWM approach, with 32.6 (offline) and 35.5 (online), so our conclusions still hold.
>
> [1] Cheng, Zhoujun, Jungo Kasai, and Tao Yu. "Batch prompting: Efficient inference with large language model apis." arXiv preprint arXiv:2301.08721 (2023).

---

> ### Comment · Reviewer_Wfe1 · 2024-12-03
> **Response to Authors**
>
> Thank you so much for your response!
>
> W2: I disagree. A lot of methods do not assume the existence and perception of tasks from the same test task distribution; i.e., they do not accumulate knowledge across different tasks from the same test task distribution. I'm not suggesting that exploiting such information is a bad *practice*, but the purpose of academic benchmarks is to simulate real-world scenarios on a smaller scale. In order to show real-world effectiveness, I believe its scalability should be demonstrated at a much larger scale, where the task distribution is broader and making connections between different test tasks is nontrivial.
>
> W1: In addition to my response regarding W2 above, my point about generalization was that we cannot always assume relevant experiences from the same test task distribution.
>
> Overall, I appreciate the authors' efforts in preparing the rebuttal, but I keep my original view on this submission.

---

### Official Review · Reviewer_NQfK · 2024-11-07

**Soundness:** 3
**Presentation:** 3
**Contribution:** 3
**Rating:** 3
**Confidence:** 4

**Summary:**

This paper introduces Agent Workflow Memory (AWM), a method for improving language model agents in complex, long-horizon tasks by learning and reusing task workflows. AWM flexibly supports both offline and online applications, showing strong improvements on web navigation benchmarks Mind2Web and WebArena. Results demonstrate notable gains in task success rates, efficiency, and cross-domain generalization.

**Strengths:**

1. Using reusable routines from agent trajectories is a fresh and effective idea for complex task management.
2. Extensive evaluations, including ablation studies, validate AWM’s effectiveness across online and offline settings.
3. The concept of workflows is well-explained in Introduction, making AWM’s approach easy to understand.

**Weaknesses:**

1. Figure 4 could be refined to better illustrate the workflow in the online setting.

2. In WebArena, if AWM leverages test set tasks to build workflows, it could cause data leakage and compromise fairness.

3. The paper does not show AWM’s sensitivity to task exploration order, which could affect its performance. For example, which task should be explored first? Task sequence could impact the quality of workflows induced—precisely, whether exploring tasks from difficult to easy (or vice versa) could degrade performance. Analyzing how task order affects workflow quality and overall results would provide valuable insight into AWM's robustness.

4. In Table 1, “# steps” is unclear. Reporting tokens or time would better reflect agent efficiency.

5. Section 5's description of “wrapping each workflow into a high-level function” needs more explanation and details.

6. In Table 4, MindAct consistently achieves a higher AF1 score than AWM. It would be helpful if the authors could explain this.

**Questions:**

AWM’s high performance in domains like Maps deserves more detailed analysis to explain these gains, which can help to understand methods.

---

> ### Author Response · Authors · 2024-11-17
>
> Thank you for recognizing the effectiveness of our method, the comprehensive experiments we conducted, and the clarity in the method description.
>
> **W3: AWM sensitivity to example order**
> It’s a good observation that example ordering may affect the AWM agent performance. We can test the effect of example ordering on the Mind2Web dataset (cross-task split) using AWM online approach. Specifically, we compare original order, random shuffling, ordering from easiest to hardest, and ordering from the hardest to the easiest (the more steps in the ground-truth trajectory, the harder the task).
> Results are shown in the table below.
>
> | Method | Element Acc. | Action F1 | Step SR | SR |
> | -- | -- | -- | -- | -- |
> | Mind2Act (baseline) | 41.6 | 60.6 | 36.2 | 2.8 |
> | Original Ordering | 50.6 | 57.3 | 45.1 | 4.8 |
> | Random Shuffle | 49.4 | 57.9 | 45.9 | 4.0 |
> | Easy-to-Hard | 49.8 | 57.8 | 45.7 | 4.0 |
> | Hard-to-Easy | 48.5 | 59.0 | 45.6 | 4.2 |
>
> First of all, AWM using any example ordering still substantially outperforms the MindAct baseline.
> Moreover, the ordering of examples does not significantly affect the performance of our AWM approach, where all four example ordering achieves similar success rates. Coupled with a careful analysis of the derived workflows, we found that our design of “sub-task level” workflow contributes to AWM’s robustness to example ordering — regardless of the complexity of the task, our method can induce useable workflow.
>
> However, for the WebArena dataset we experimented on, the examples are intentionally ordered in a specific ordering to maintain the validity of the browser environment. Therefore, we could not change the orders arbitrarily to examine this effect.
>
>
> **W4: “# steps” is unclear**
> We argue that “# steps” is more informative than “# tokens” in agentic settings, where each step represents an agent action such as “click(12)”; where tokens may mislead this number, e.g., a step “click(12)” and a step “send_msg_to_user(“The highest score is 98”)” takes roughly same amount of time and same number of LM calls; however, if presented by tokens, “click(12)” has four tokens [click, (, 12, )] and “send_msg_to_user(“The highest score is 98”)” has 13 tokens [send, _msg, _to, _user, (, “, The, highest, score, is, 98, “, )], which misleadingly shows a big difference.
> Moreover, the number of actions/steps is a commonly adopted metric in agent works, e.g., SteP [1].
>
> W5: Explain “wrapping each workflow into a high-level function”
> We provide concrete examples for wrapping workflows into high-level functions in line 480–482, where we rewrite the textual workflows to executable functions, and the step in textual workflows turns to lines of programs in the high-level function. More concretely, given a textual workflow:
> ```
> Workflow: login to the website
> # I will first click the username box to input
> click(box1-id)
> # next, type in the username
> type(box1-id, username)
> # after finishing the username, I will now click the password textbox
> click(box2-id)
> # then, input the password
> type(box2-id, password)
> # lastly, click the submit button to initiate the login process
> click(submit-id)
> ```
>  We turn it into a high-level function:
> ```python
> def login_to_website(username, password, box1-id, box2-id):
>     click(box1-id)
>     type(box1-id, username)
>     click(box2-id)
>     type(box2-id, password)
>     click(submit-id)
> ```
>
> **Q1: Explaining high gains on Maps Website**
> In short, compared to the other websites, the tasks on Maps website are relatively simple and similar to each other, thus benefiting more from our workflow reuse strategy.
>
> We provide a detailed illustration of the agent learning process on the Maps website in Figure 1, where all marked workflows and accumulative success rates are accurately depicted. As we stated in section 3.1.3, validated with quantitative experiment results in Table 2, and illustrated the major pattern of AWM in Figure 6, our AWM-equipped agent learns increasingly complex workflows built on top of earlier, easier ones, thus being able to solve more and harder problems over time, leading to higher results than baselines. We do not modify our method in any way particularly for the Maps website.
>
> **W1: Refine Figure 4**
> Thank you for pointing this out. To possibly improve the figure, we could (1) add arrows to indicate example ordering and streaming, (2) use icon to highlight the “induce” part. Free feel to provide more suggestions and we are happy to incorporate them.
>
> **W2: Causing WebArena data leakage**
> AWM does not leak WebArena data. As we highlighted in Section 1, 2, and Figure 4, our method **only requires test queries**, and never uses ground-truth trajectories nor evaluations. Since our AWM agent never has access to any potential “answers”, we believe our method will not cause WebArena data leakage.
>
> [1] Sodhi, Paloma, et al. "Step: Stacked llm policies for web actions." First Conference on Language Modeling. 2024.

---

> > ### Comment · Reviewer_NQfK · 2024-12-02
> >
> > W3: AWM sensitivity to example order
> > Thanks for the additional experiments on Mind2Web. However, I believe testing example ordering on WebArena is more critical, as some tasks may act as prerequisites or demonstrations for harder ones.
> >
> > W4: “# steps” is unclear
> > Thanks for the clarification. While steps are informative, reporting both steps and tokens as optional metrics could provide a more comprehensive view.
> >
> > W2: Causing WebArena data leakage
> > Appreciate the clarification. However, see my comment on W3 regarding the potential importance of task ordering and evaluation methodology.
> >
> > General feedback
> > Thanks for addressing other concerns clearly. Enhancing Figure 4 as suggested will improve understanding.

---

### Official Review · Reviewer_NWwU · 2024-11-07

**Soundness:** 2
**Presentation:** 3
**Contribution:** 2
**Rating:** 6
**Confidence:** 3

**Summary:**

The paper presents the Agent Workflow Memory method for extracting and using task workflows for web navigation task.
The proposed memory is a collection of subroutines each of which is comprised from a task description and a set of actions generated by the agent or taken from the training dataset.

AWM is implemented with GPT-3.5 and GPT-4 and evaluated on WebArena and Mind2Web datasets.

**Strengths:**

- The proposed method enables online and offline variants of generating the task workflows to use both high-quality train data and past experiences of the agent.
- The authors conduct the ablation study and evaluations on two web navigation datasets.
- The paper is well-written and easy to follow.

**Weaknesses:**

- The paper would benefit from the descriptions of how the agent was implemented and tested. The authors provide the model prompt and example workflow prompts in appendix, but considering that the agent is based on the closed-source models, the lack of implementation details impedes reproducibility.
- It is not clear how the workflows are extracted from memory to improve the task solving or the whole memory content is always provided as a part of the prompt.

**Questions:**

- How many past experiences can be stored in memory?

---

> ### Author Response · Authors · 2024-11-17
>
> Thank you for recognizing the effectiveness of our method, careful experiments, and clarity in paper writing.
>
> **W1: Add description for implementation details of closed-source models**
> We agree that closed-sourced models such as the GPT models provide limited explainability, nonetheless, interfacing with closed-source modes is common, as in many impactful works and importantly most related works. Moreover, we want to show that our AWM approach is compatible with and improves state-of-the-art models, which are all closed-source.
> To provide more implementation details for our AWM method, we described all hyper-parameter settings in lines 214–215 (for webarena) and lines 317–323 (for mind2web). We also provided our entire codebase in the supplementary material, which includes all implementation details of our method.
>
>
> **W2: Not clear how memory is extracted/used**
> As illustrated in Figure2 and explained in section 2.3, we induce workflows from past experiences then add them to the agent memory (Figure 2, step 3) – we input (NL, trajectory) examples to the model and prompt it (see prompt in Appendix A.1) to output workflows (see examples in Appendix A.2); we add workflows to model contexts at next action generation process.
> If you are asking how the database of all workflows is used to solve a new problem — we add all workflows in the database into the model inputs. The database as 7.3/7.4 workflows on average for WebArena/Mind2Web, so can be easily fitted into model context.
>
> **Q1: How many workflows can be stored in memory**
> We do not limit the number of workflows stored in the memory. Empirically, our agent uses 7.4 workflows per website on WebArena, and 7.3 workflows per website for Mind2Web, as presented in Table 10.

---

> > ### Comment · Reviewer_NWwU · 2024-12-02
> >
> > Thank you for your answers. My questions and concerns were addressed and I have raised the score accordingly.

---

### Official Review · Reviewer_1vxx · 2024-11-27

**Soundness:** 2
**Presentation:** 3
**Contribution:** 2
**Rating:** 5
**Confidence:** 3

**Summary:**

Paper describes an approach for agents to learn from past experiences using the notion of reusable task workflows and adapt to complex tasks. Overall, the paper is well written. However, establishing the novelty of the proposed method. By providing a clear differentiation from existing work would make this contribution’s impact more stronger.

**Strengths:**

- Very flexible method which can sustain both offline and online cases.
- Strong improvement over benchmarks even those containing expertly labeled examples.
- Examination of workflow quality along with ablations is well written

**Weaknesses:**

- The memory integration part is sub-optimal, where workflow integration is a simple addition. Using a more human-like mechanism such as attention based memory access or some sort of a weighted retrieval would be better.
- As I was reading the paper, hierarchical reinforcement learning came to mind. Having some analysis on how AWM compares with learning hierarchical policies implicitly would be helpful.

**Questions:**

- How does AWM handle situations where the induced workflows are incorrect or suboptimal?
- Why bother grouping actions into workflows instead of simply learning which action to take in each state?

---

### Meta-Review · Area_Chair_kNZL · 2024-12-23

**Metareview:**

This paper develops an approach to summarize interactions with past web-agent-like environments to improve the performance on a given prompt. The essential approach consists of converting a successful trajectory (either obtained by the agent, or from the dataset) into a “workflow” for each website. For each website, the agent retrieves these workflows into its working memory for inference.

Based on comments of the reviewers, the authors have made a number of improvements to the manuscript during the rebuttal phase. The key concerns that are yet to be addressed however pertain to (i) clarity of the narrative that describes the approach with sufficient details and rigor, and (ii) ascertaining whether the approach unduly benefits from the correlations of tasks used to create workflows with new test tasks (and evaluating against appropriate baselines in this setting). Regarding the second point, it would be interesting if this approach can use workflows that are shared across websites and control for the number (or length) of these workflows explicitly. Regarding the first point, it is also important to use absolute success rate (as opposed to “relative improvement in the success rate” as the paper does at many places). The authors are also encouraged to compute error bars (e.g., via bootstrap) for the numerical results in the tables to ensure statistical significance of these results.

**Additional Comments On Reviewer Discussion:**

Reviewer NWwU wanted more details of the implementation of the approach. The authors have provided these details in the rebuttal.

Reviewer NQfK wanted to clarify whether the approach in this paper uses tasks from the test set to create the workflows, and whether the workflow memory would be affected by reordering tasks differently. The authors have partially addressed these issues with more experiments.

Reviewer Wfe1 was concerned about (i) generalization of workflows (since they are specific to each website), and (ii) the correlations between previous tasks that are used to create workflows and new test tasks that may be responsible for the improved performance. The authors have demonstrated some evidence of cross-website generalization. But the reviewer was not satisfied with the response to the second point.

Reviewer fnjL pointed out that the writing of the paper was not clear (I share this opinion). They wanted some clarifications on how the workflows are computed, comparison with state of the art methods while controlling for inference-time computation and a discussion on the limitations. The authors have partially addressed these comments in their rebuttal. The reviewer is not convinced about whether the paper has done a thorough job of exploring the proposed approach.

---

### Decision · Program_Chairs · 2025-01-22

Reject